# Effects of Seeding Pattern and Cultivar on Productivity of Baby Spinach (*Spinacia oleracea*) Grown Hydroponically in Deep-Water Culture

**Daniel B. Janeczko and Michael B. Timmons ***

Department of Biological and Environmental Engineering, Cornell University, Ithaca, NY 14853, USA; dbj42@cornell.edu

* Correspondence: mbt3@cornell.edu

**Abstract:** Baby spinach (*Spinacia oleracea*) was grown in a bench-scale deep-water culture (DWC) system in expanded polystyrene (EPS) plug trays. Two experiments were performed. In the first, different seeding patterns, [1-2-1-2 . . . ] or [3-0-3-0 . . . ] seeds per sequential cell, at the same overall density per tray, were compared to evaluate the potential of an EPS tray designed with fewer cells, but sown with more seeds per cell (to preserve canopy density). Using such a flat would lower growing substrate requirements. Seeding in the [3-0-3-0 . . . ] pattern reduced seed germination, but only by 5%. Harvested fresh weight was also less numerically in the [3-0-3-0 . . . ] pattern but not statistically. The second experiment observed cultivars Carmel, Seaside and Space grown concurrently. Carmel had the highest germination, nearly 100%, which was significantly greater than Seaside but not Space. Germination for Space was not significantly different from that of Seaside. Carmel also had the highest harvested fresh weight but was not significantly different from Space; both Carmel and Space produced significantly more harvested fresh weight than Seaside.

**Keywords:** hydroponics; deep-water culture; baby spinach; Carmel; Seaside; pericarp; dibbler; density; *Spinacia oleracea*

## 1. Introduction

Hydroponics is the soilless culture of plants grown in a nutrient solution. Within hydroponics, several different techniques are employed to expose plant roots to a nutrient solution. The Deep-Water Culture (DWC) method is characterized by use of a trough or tub filled with aerated nutrient solution to which a floating raft is added that holds the plants with their roots suspended vertically in the water column [1].

Hydroponic greenhouse crop production has the potential to reduce dependence on imported crops and to increase food security and safety while simultaneously enhancing environmental quality and energy/resource efficiency of food production systems [2,3]. Baby spinach is an attractive choice for hydroponic production since it is popular as a fresh vegetable and has relatively high concentrations of bioactive compounds including ascorbic acid, carotenoids, and flavonoids that contribute to its high nutritional value, despite its low caloric content [4]. Growing this crop in the northeast is particularly attractive since 86% of US fresh spinach production in 2017 (53,030 acres total) occurred in Arizona or California [5].

Expanded polystyrene (EPS) flats with varying cell volume and spacing are commonly used for baby leaf production in DWC systems. Typical cell volumes can range from 2 or 3 cm$^3$ to 10 cm$^3$. Cell volume impacted tomato, eggplant, and pepper seedling fresh weight when grown from seed [6]. Planting (canopy) density was less predictive of seedling fresh weight (within the range considered:

94–1125 plants/m$^2$) [6]. Baby romaine lettuce (*Lactuca sativa*) seeded at higher densities in EPS flats had greater yield per unit area and resulted in plants with longer but fewer leaves compared to lower-density arrangements [7]. This demonstrated that planting density and available cell volume can impact the morphological development of baby leaf crops. However, the effect of proportional cell volume available to spinach seedlings and the higher density of plants in the early stages of growth associated with sowing multiple seeds per cell have not been thoroughly investigated.

The soilless medium in which seeds are sown is often a large fixed material cost of each crop. When sowing an off-the-shelf EPS flat (generally re-used for 30 successive plantings), with sphagnum peat moss germination mix and spinach seed at a density of 1.5 seeds per cell, the proportional cost of growing medium can be estimated to be over 50% of the total material input costs (Figure 1). Employing a flat with fewer cells, but sowing more seeds in each cell (thus preserving the same canopy density), could meaningfully reduce material expenses. While the density of the canopy would theoretically remain unchanged, it is unclear how packing more seeds into each cell affects germination and growth throughout the short crop cycle. It is hypothesized that seeding more seeds per cell will negatively affect germination and growth as the seed-substrate contact needed for seeds, especially large ones, to imbibe and germinate will be disrupted in cells with multiple seeds by the earliest germinating seeds [8]. One of our objectives was to evaluate this hypothesis.

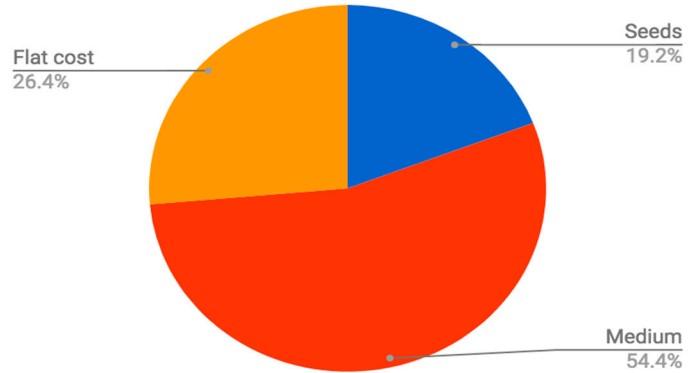

**Figure 1.** Flat material input costs.

Spinach is not considered to be a high-value crop by producers and generally not tailored to hydroponic production. A main driving force behind cultivar selection during seed production is resistance to blue mold aimed at field-grown applications, but this is not of critical importance for hydroponic baby leaf production when compared to other cultivar characteristics such as simultaneity of germination, germination rate, and plant morphology during the short crop cycle [9]. Recent research on hydroponic baby leaf spinach production in DWC has focused on two round-leaf, downy mildew-resistant cultivars: Space and Carmel. Seaside is a smooth-leaf, slow-bolting cultivar that has not been thoroughly tested in a DWC system [2,10]. Comparative testing of these cultivars is a next step in determining which will grow vigorously in a DWC environment.

The objectives of this study were: a) to determine effects of seeding methods on performance of baby spinach planted at high densities and b) to compare performance characteristics from seeding to harvest for three cultivars (Carmel, Seaside, and Space) using one of the high density seeding methods.

## 2. Materials and Methods

Experiments were conducted to test seeding patterns (Experiment 1) and performance characteristics (Experiment 2). Germination, fresh weight production, and observations of canopy characteristics were considered the most relevant metrics to evaluate treatments. Experiment 1 compared seeding patterns with the Carmel variety only and consisted of either seeding identically sized cells with a one-two-one alternating pattern [1-2-1-2] of seeds per sequential cell or using three seeds in a cell and then an empty cell and then three seeds in a cell etc. [3-0-3-0], giving the same

average density of 1.5 seeds per cell in both treatments. Experiment 2 observed three cultivars that were grown concurrently side by side, using the cultivars Carmel, Seaside, and Space seeded at a fixed density of 1.5 seeds per cell in the [1-2-1-2] pattern. Data on both germination and fresh weight was collected by counting seedlings in each treatment mid-way through the crop cycle and weighing hand-harvested leaves after they reached marketable size.

Speedling (Ruskin, FL) 338 cell flats were used for all experiments (see Figures 3 and 9a for a representative flat). The flats had dimensions of 34.3 cm by 57.2 cm by 4.4 cm. Each flat was divided into two blocks by using two center rows (no seeds) to separate blocks. Thus, each block had 130 cells for planting. Both seeding patterns resulted in 195 seeds being planted per block with a potential to have 195 seedlings. Treatments were assigned randomly to the blocks. Cells are square on the top, 4.4 cm deep, spaced on 2.54 cm centers, and tapered to a square open bottom, giving the cells a pyramid-like shape. The blocks described above are replications of the treatments imposed in both experiments. An advantage of this approach was that time or season was eliminated as a variable that could impact response in an individual experiment.

Experiments took place in Ithaca, NY (42∘26056.2″ N 76∘28008.3″ W). Flats were filled and seeded, then placed in a dark, climate-controlled growth chamber where germination occurred for two and a half days after seeding. Subsequently, flats were moved to a conventional glass greenhouse and transferred into tubs using a bench-scale deep-water culture (DWC) system, where they remained until harvest. A summary of seeding, floating, and harvest dates are given in Table 1.

**Table 1.** Experiment descriptions and dates.

| Experiment Number | Experiment Description | Seeding Date | Float Date | Harvest Date |
|:---:|:---:|:---:|:---:|:---:|
| 1 | Seeding Pattern | 10/11/2017 | 10/13/2017 | 10/28/2017 |
| 2 | Cultivar Comparison | 10/28/2017 | 10/30/2017 | 11/17/2017 |

### 2.1. Germination Chamber and Greenhouse Description

A climate controlled chamber ($0.93 \text{ m}^2$) with no light and constant temperature of 24 °C was used for germination. The DWC system was located on a waist-high bench in the NE corner of a conventional glass greenhouse. The greenhouse thermal characteristics are well-described by Vandam et al. [10] who raised spinach in a greenhouse section within the same complex under similar operational protocols. No supplemental light was provided.

### 2.2. Greenhouse Environmental Conditions

Greenhouse ambient temperature was maintained at 24 °C from 8 am–8 pm and 19 °C at night by an automated Argus system (Argus Control Systems Ltd., Surrey, BC, Canada). The relatively new glass structure was estimated to have a solar light transmissivity of 65% based upon previous measurements by our research group.

Light values were calculated using $W/m^2$ readings gathered from an outdoor sensor connected to the greenhouse automation system that controlled the shade curtain and took readings at ten minute intervals. The quantum units were converted to PAR and multiplied by the transmissivity of the structure (65%). When the curtain was drawn over the bench on any time step, the PAR value was multiplied by the transmissivity of the shade curtain (40%) to estimate PAR reaching the canopy. These PAR values were integrated over 24 hour periods, yielding daily light integral (DLI) values.

For Experiment 1 (seeding pattern) in which the flats were in the DWC tubs for 16 days, the average cumulative light received was $5.5 \text{ mol/m}^2/\text{day}$ (Std. Dev. = 1.46) for a total of 88.4 mol per $m^2$ over the entire crop cycle. For Experiment 2 (cultivar comparison), the average light received was $3.0 \text{ mol/m}^2/\text{day}$ (Std. Dev. = 1.72) for a total of 56.2 mol per $m^2$ received over the 19-day crop cycle. In both experiments the light levels were below the $17 \text{ mol/m}^2/\text{day}$ considered ideal for continuous production [11].

### 2.3. Deep-Water Culture System Description

The production period for the spinach plants was conducted using two steel tubs (61 cm by 122 cm by 30.5 cm deep) which allowed five cut flats (described below) to be floated when completely utilized. Nutrient solution (described in Section 2.3) flowed freely between the two channels through a 1.9 cm ID hose. Tubs were insulated with 1.9 cm polystyrene boards on their exterior to minimize heat transfer and reduce cooling load. A 1/6 HP submersible pump was used to pump nutrient water through a venturi injector (for aeration), followed by a $\frac{1}{4}$ HP inline chiller that kept the water temperature at 19 °C. Water was returned to the tubs through a 2.5 cm ID PVC pipe with outlets into each tub. The nutrient water and root zone were maintained below 20 °C and highly aerated, since this temperature has been to be effective for mitigating spread of *Pythium* infection, especially *P. aphanadermatum*, and the high degree of aeration also contributes to greater resistance to infection [12]. Oxygen levels were also enhanced by using two air stones in each tub. Dissolved oxygen saturation was verified using a calibrated YSI Pro20 (Yellow Springs, OH, USA) meter when the system was operating.

Sections of the tubs that were not occupied by flats were covered with Styrofoam™ to prevent growth of algae and contaminants entering the water. Between experiments, the tubs were emptied, cleaned, and sanitized with Greenshield (BASF, Ludwigshafen, Germany) to minimize risk of root infection and ensure consistent nutrient solution conditions between experiments. Figure 2 shows the experimental setup and placement of tubs, chiller and air pump.

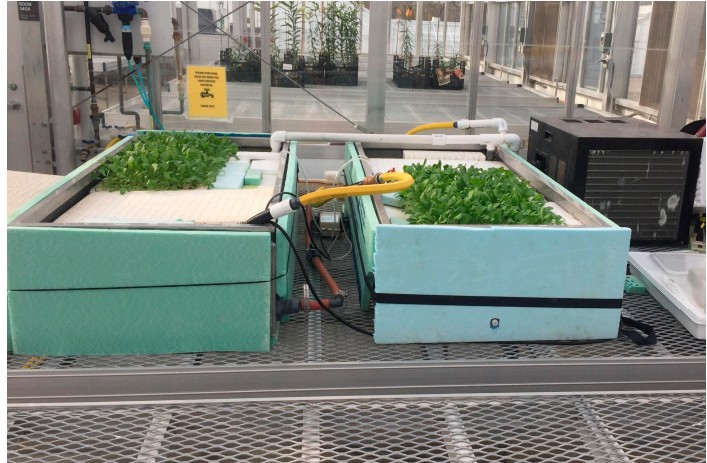

**Figure 2.** Bench-scale deep-water culture (DWC) system at half-capacity.

### 2.4. Nutrient Solution Conditions

The stock solutions recommended by Sonneveld and Straver [13] were prepared per the recipe (Table 2). The day before the flats were removed from the germination chamber for floating, stock solutions were added to reverse osmosis (RO) water in the tubs in a 1:1 ratio (A:B) to achieve an electrical conductivity (EC) of 1200–1400 µS/cm. This corresponds to a half strength solution as recommended by Sonneveld and Straver [13]. The half strength solution has been shown to be effective for lettuce and spinach [10,14]. The nutrient solution resulted in a pH between 5.7 and 6.3 without the addition of any additional acid or base over the course of each experiment.

Throughout each experiment's growth period, RO water and stock solutions were added to maintain the EC and pH within the desired range as checked by handheld digital meters, calibrated at the beginning of every experiment and every two days thereafter. Water level in the tubs was maintained high enough so that growing flats were not shaded by the tub walls.

**Table 2.** Stock solution recipes.

| Stock A—Chemicals/salts added to 30 L of RO Water | |
| --- | --- |
| Calcium Nitrate | 2916.0 g |
| Potassium Nitrate | 613.2 g |
| Ammonium Nitrate | 84.0 g |
| Sprint 330 Iron—DTPA (10% Iron) | 67.0 g |
| **Stock B—Chemicals/salts added to 30 L Water** | |
| Potassium Nitrate | 2037.8 g |
| Monopotassium Phosphate | 816.0 g |
| Potassium Sulfate | 65.6 g |
| Magnesium Sulfate | 1478.0 g |
| Manganese Sulfate* $H_2O$ (25%Mn) | 2.56 g |
| Zinc Sulfate* $H_2O$ (35%Zn) | 3.44 g |
| Boric Acid | 5.58 g |
| Copper Sulfate* $5H_2O$ (25%Cu) | 0.56 g |
| Ammonium Molybdate | 0.26 g |

## 2.5. Dibbling Tools

A combination of conventionally built and 3D-printed dibbling tools was used for these experiments (Figures 3 and 4). Dibblers developed by Cornell CEA specifically for spinach production create a divot in which seeds are sown, atop which more substrate can be added so that the large spinach seeds were adequately deep and covered in the EPS flats.

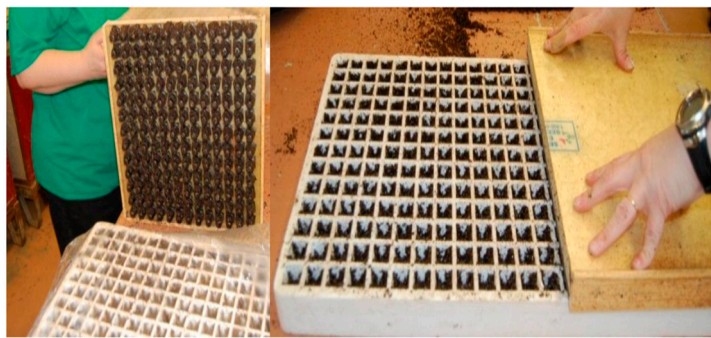

**Figure 3.** Conventionally-built plate dibbler.

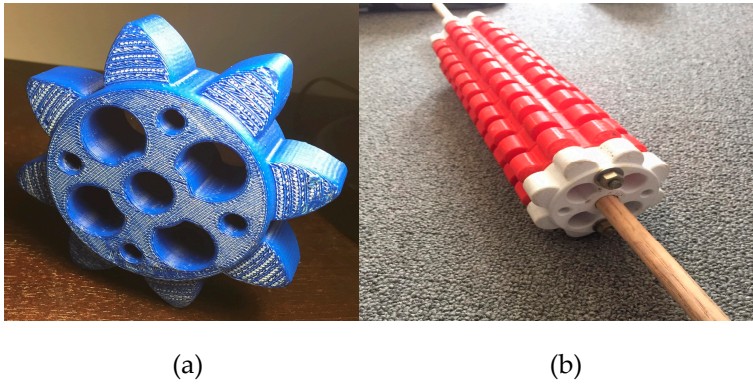

(a)　　　　　　　　　　　　　　　　　(b)

**Figure 4.** (**a**) Single rolling dibbler; (**b**) Whole flat dibble roller.

## 2.6. Seeds

Seeds were purchased from Jonny's Seeds Inc. (Winslow, ME) in September 2017. When not in use, they were stored in a cool, dark, dry environment. Germination percentages reported by the

supplier were: Carmel—84%, Seaside— 97%, Space—99%. Carmel and Space are both semi-savoy, meaning the true leaves have a slightly crinkled appearance. Seaside is a smooth-leaf variety.

### 2.7. Sowing, Germination, and Floating Procedure

Before seeding, flats were cleaned, disinfected with a Greenshield solution, thoroughly rinsed, and dried. Sungro (Vancouver, BC, Canada) Propagation mix, composed of sphagnum peat moss, horticultural vermiculite, pH adjustment lime, wetting agent, and starter nutrients was used to fill flats. Reverse Osmosis (RO) water was added to the dry medium at a ratio of 0.6 kg RO water per 1 kg dry medium and mixed in 20 L buckets. This moisture content was chosen from a trial run that showed comparable performance between flats germinated with either 60% or 100% medium wet weight to medium dry weight. The medium and water were well-mixed, and the lid was kept on the buckets except when mixing or filling flats to minimize moisture loss.

When filling the flats for the first pass, a 0.6 cm metal mesh screen (Figure 5) was used as a sieve to remove any clumps from the medium and ensure homogenous, loosely-packed filling of cells. Excess medium was scraped off the top of the flat with a straight edge. The plate dibbler in Figure 3 was used to create consistent dibbles in each cell and then flats were seeded at the appropriate density using the designated cultivar.

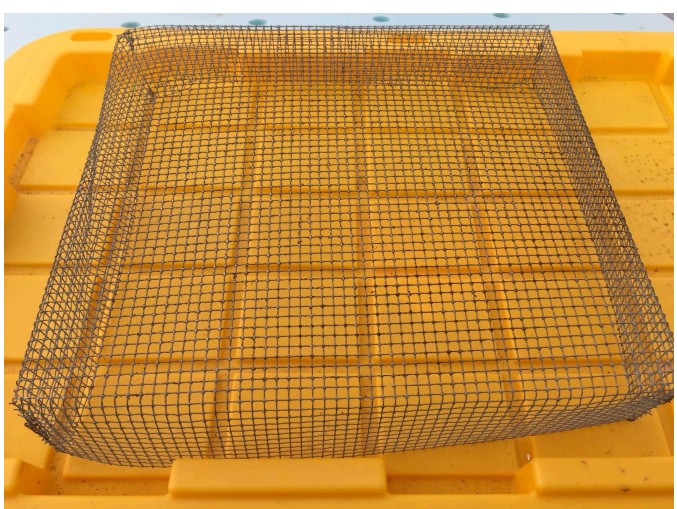

**Figure 5.** Metal mesh sieve used to remove clumps and ensure loose filling.

After initial seeding, another layer of medium was added on top of the seeds, following the same sieving procedure described for the first pass. Finally, the rolling dibbler pictured in Figure 4 was used to compress the medium on top of the seeds in the second pass by rolling the dibbler across the flat surface back and forth several times.

All flats were stacked randomly after they had been seeded, filled, and dibbled. Non-seeded guard flats were placed on the top and bottom of the stack. The entire stack of flats was wrapped in plastic to prevent moisture loss. The stack was moved into a larger, rigid plastic bin for transportation to the germination chamber, where the flats were stored for 60 hours in the dark at 24 °C before removal for inspection and floating.

Upon removal from the germination chamber, flats were inspected for pests and abnormalities. Throughout the experiments, no pests were found in any seedlings. Germination appeared successful upon removal from the chamber, meaning there were some visible seedlings protruding from the medium, as seen in Figure 6, and many more young roots protruding out of the bottoms of flats. After this step, the flats were floated in the tubs, where they remained until removal for harvest.

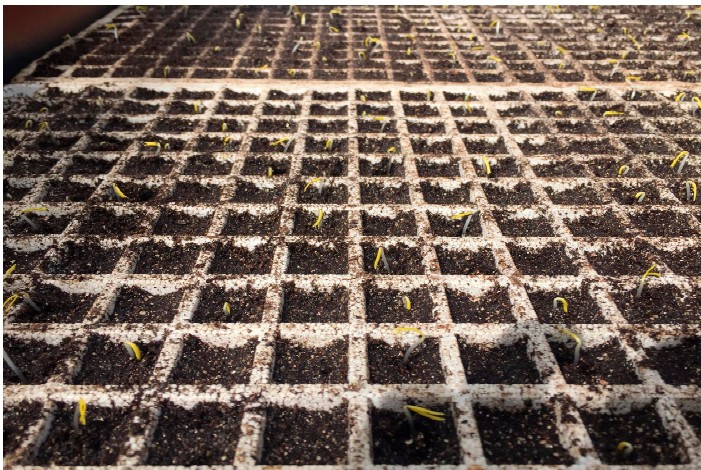

**Figure 6.** Flats just after floating with seedlings breaching the medium surface.

*2.8. Seedling Count*

On the sixth day after floating in the tubs, the visible seedlings were counted for each treatment. This count included "popups" (seedlings that withered because their roots did not reach the water), which were uncommon [9]. Counting seedlings earlier than six days was determined to be a poor indication of germination and initial growth success.

*2.9. Harvest Protocol*

Flats were harvested when the fastest growing leaves of a treatment reached marketable size, which occurred between 14 and 18 days after transfer to the tubs. This harvest date initiation was chosen so that no leaves exceeded 7.5 cm in length (when measured from the base of the leaf).

Harvest was performed by hand with scissors. A handful of leaves would be carefully grasped. The stems would then be cut about halfway down the stems of the longest leaves. This meant some smaller leaves sitting lower in the canopy were harvested as well, but with shorter stems. A large variance in leaf size was found for all treatments, which was due both to the non-simultaneous germination observed, and the presence of young, second true leaves in addition to the larger first true leaves. Harvest weight was measured immediately after leaves were harvested using a digital scale accurate to one gram. As can be seen in Figure 7, many small leaves sitting low in the canopy were not harvested.

*2.10. Statistical Analysis*

In Experiment 1 (seeding pattern with single cultivar) there were four replications of each pattern and in Experiment 2 (cultivar response to fixed seeding method), there were four replications of the Carmel and Seaside cultivars and two replications of the Space cultivar. For both experiments, a one-way analysis of variance using Tukey–Kramer paired t-tests was performed using JMP PRO 11 statistical analysis software [15]. The average and standard deviation of each treatment (pattern or cultivar) except for the cultivar Space was calculated for both final seedling counts and fresh weight. Data from a typical high-germinating flat is presented to show the intra-flat variation observed, even when germination was high.

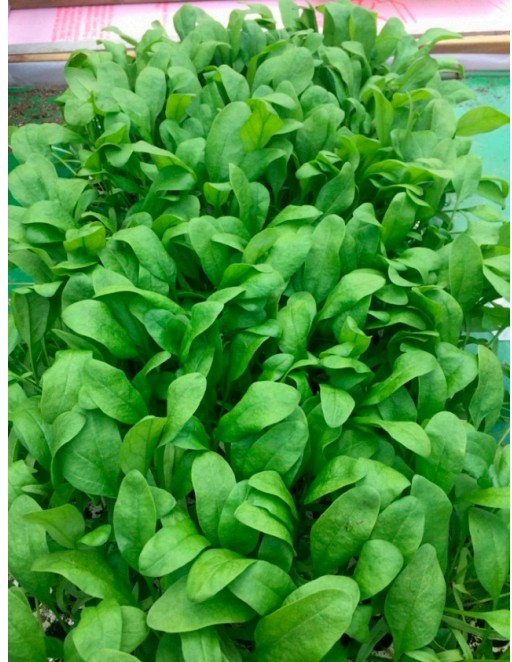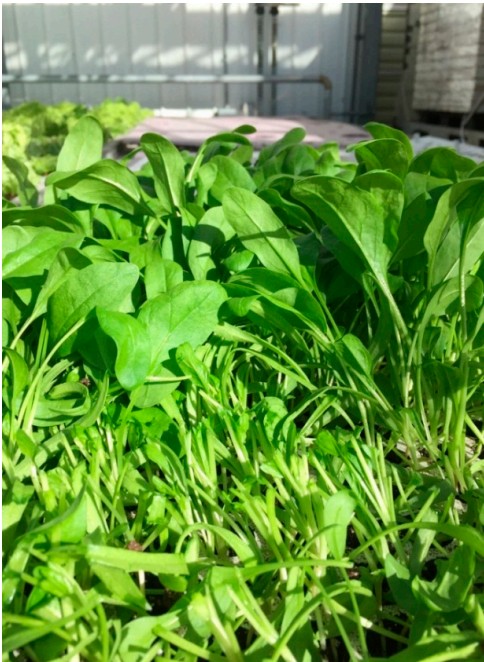

**Figure 7.** Before and mid-harvest of Carmel; some small second true leaves are left behind low in the canopy.

## 3. Results

### 3.1. Experiment 1: Seeding Pattern

Results from the seeding pattern experiment showed that seeding pattern [1-2-1-2] or [3-0-3-0] negatively affected germination. There were 157 (Std. Dev. = 4.2) seedlings per replicate block in the [1-2-1-2] pattern versus 148 (Std. Dev. = 4.5) seedlings per replicate block in the [3-0-3-0] pattern (Table 3) from a total of 195 seeds planted per replicate block. The Carmel cultivar had a manufacturer reported sprouting percentage of 83%. Our results for both treatments achieved high percentages of the manufacturer's reported germination percentage although numerically the [1-2-1-2] pattern was 97% of the reported value while the [3-0-3-0] pattern was 91% of the reported value. Fresh weight harvest yield was lower numerically but not statistically with the [3-0-3-0] sowing pattern as well. Since the reductions in germination and yield were less than 10%, growers should consider and evaluate the cost of additional substrate versus reductions in performance based upon these results.

**Table 3.** Effects of seeding pattern on seed germination from 195 seeds per replicate block and harvested fresh weight yield of Carmel spinach. Cells were seeded according in [1-2-1-2] or [3-0-3-0] patterns giving average density of 1.5 seeds per cell. Different superscripts indicate significant differences between treatments by Tukey–Kramer paired *t*-tests at α = 5%.

| Treatment | Replication | Seedling Count | Fresh Weight (g) |
|---|---|---|---|
| [1-2-1-2] | 1 | 150 | 144 |
| | 2 | 159 | 157 |
| | 3 | 158 | 148 |
| | 4 | 161 | 156 |
| | Mean | 157 [a] | 151.2 [a] |
| | Std. Dev. | 4.2 | 5.4 |
| [3-0-3-0] | 1 | 144 | 134 |
| | 2 | 155 | 152 |
| | 3 | 144 | 146 |
| | 4 | 149 | 145 |
| | Mean | 148 [b] | 144.2 [a] |
| | Std. Dev. | 4.5 | 6.5 |

### 3.2. Cultivar Comparison

As shown in Table 4, Carmel and Space cultivars had a significantly higher germination percentage than Seaside ($\alpha$ = 5%) and fresh weight yield production ($\alpha$ = 5%). Carmel had the highest germination (81%) of the three cultivars tested. Carmel averaged 98% of its reported 83% germination rate. Final germination percentage was stable for Carmel across both experiments; Experiment 1 was also 81%. Space and Seaside cultivars germinated and produced seedlings at a relatively high rate, as well, but at significantly lower rates than reported by the supplier.

**Table 4.** Effects of Carmel, Seaside, and Space cultivars on germination and harvested fresh weight yield with all cells seeded in a [1, 2, 1, 2] seeding pattern (195 seeds planted). Different superscripts indicate significant differences between treatments by Tukey–Kramer paired *t*-tests at $\alpha$ = 5%.

| Cultivar | Reported Germination % | Actual Germination % | Seedling Count | Harvested Fresh Weight (g) |
|---|---|---|---|---|
| Carmel | 83 | 81 | 157 [a] | 150 [a] |
| Seaside | 97 | 74 | 144 [b] | 94 [b] |
| Space | 99 | 76 | 147 [a] | 146 [a] |

Seaside baby spinach grown in the low-light conditions of November in Ithaca produced a short, bushy canopy compared to that of Carmel and Space as can be seen in Figure 8. The latter two cultivars produced longer stemmed leaves that sat higher above the cotyledons and were easier to harvest.

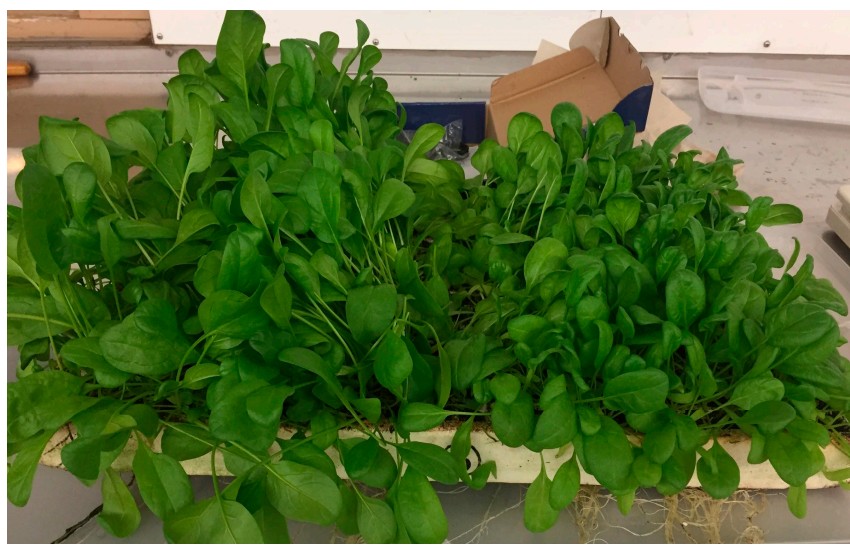

**Figure 8.** Carmel (**left**) and Seaside (**right**) at harvest illustrating the contrast between the long stems of Carmel and the short, bushy canopy of Seaside.

Pericarps are the hard outer coat of spinach seeds that are sometimes stuck to the tips of emerging cotyledons (Figure 9a). At time of harvest, they were infrequent and hard to spot as cotyledons become tangled in the lower leaves. However, it was rare to spot a pericarp on cotyledons after a week in the tubs as most were pushed off by the growing cotyledons and true leaves. (Figure 9b).

In the Carmel and Space treatments, fewer cotyledons ended up in the harvested spinach, in general, due to the longer stems of the true leaves and the nature of the cotyledons to fold over under the true leaves compared to Seaside. Overall, this resulted in almost no pericarps in the harvested leaves.

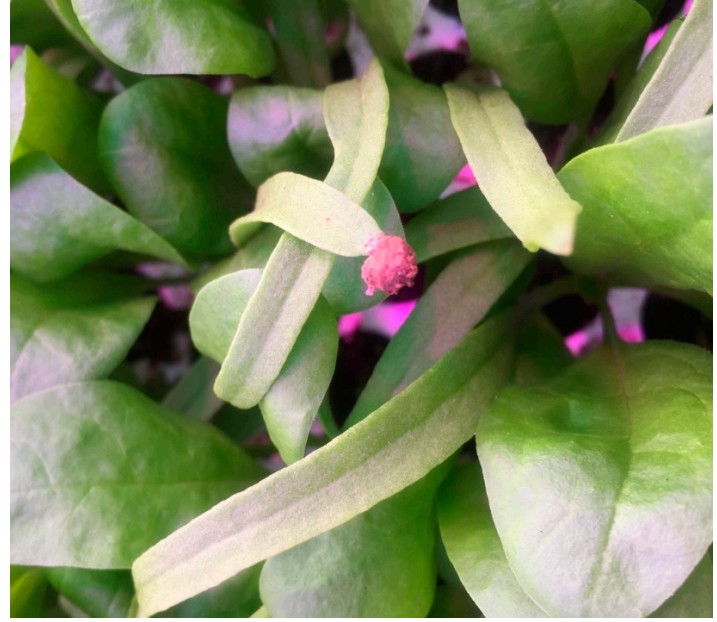

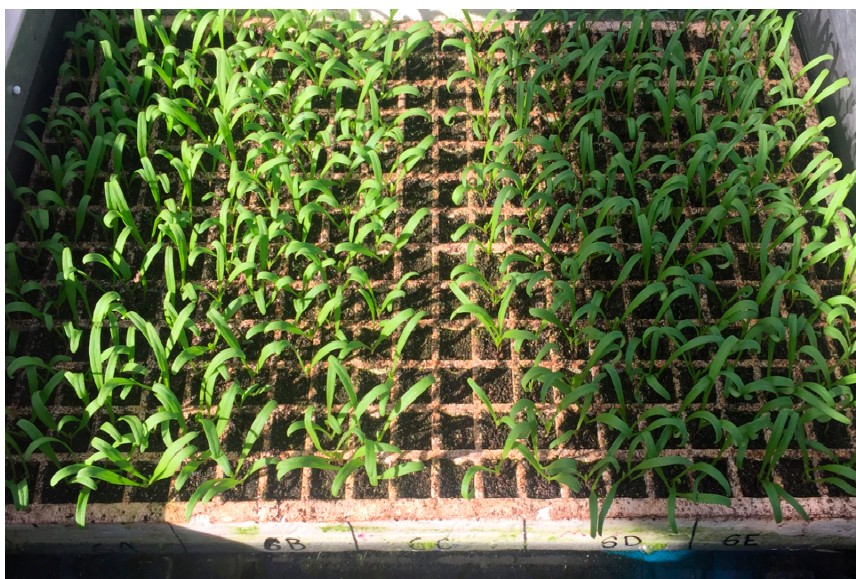

**Figure 9.** (**top**) Pericarp stuck on cotyledon late in the crop cycle; (**bottom**) A flat one week after floating with few pericarps.

### 3.3. Intra-Flat Variation

One flat of Carmel was evaluated to look at row to row variation in germination. Even for a high-germinating flat of Carmel, the variation in seeds that germinated between rows of a flat was surprising: across 22 rows, germination was 80.4% (Std. Dev. = 2.32), and ranged from 56% to 97%. As the seedlings grew, they took advantage of all available space to create a relatively uniform canopy which seemed to mitigate effects caused by intra-row variation in seed germination.

## 4. Discussion

These experiments correspond to first steps in a comprehensive comparison of the cultivars represented in these experiments. Ideally, the study would have been conducted in a growth chamber with consistent DLI, temperature, and relative humidity as in [2]. However, the germination results were consistent and largely independent of fluctuations in greenhouse light conditions. It is peculiar that the cultivar with the lowest reported germination, Carmel, had the highest overall germination

percentage (nearly 100% of that reported by the supplier) and the most vigorous growth during the short 15–18 day in-tub cycles used in our experiments. Space also performed relatively well when compared to Seaside. Seaside was the least suitable for DWC hydroponic production given its lower germination, short stems, and reduced growth.

It appears the current sowing and germination procedure is sub-optimal for Space and Seaside, which germinated less frequently than their reported rate. In future tests, cells will be taken apart and studied where no seedlings emerged to observe the seed condition. If seeds had germinated in the cell but then had not developed properly, this would suggest altering the planting procedures. For instance, if the seed germinated but the seedling withered and died before it could breach the medium surface or its roots reached the nutrient solution, different cell dimensions, seed positions in cells, or compaction/dibbling schemes should be tested. On the other hand, if cells were found to have non-germinated seeds, then different germination temperatures, a wider range of medium moisture contents, or duration spent in the germination chamber should be investigated, among other germination procedure alterations.

As mentioned previously, seedling numbers earlier than six days after floating started were not necessarily indicative of the final numbers. This suggests methods to imbibe or prime seeds could be applied to synchronize germination timing and speed up the germination of the slowest seeds. If the last seedlings are emerging several days after the first, there is a clear loss of productivity and the maximum potential of the seeds is not being realized. Cornell CEA has done extensive work on imbibing procedures [9,16,17].

The pattern experiment results showed that there was a small reduction (10%) in final yield of Carmel when fewer cells were seeded but at a higher density per cell. This implies that proportional cell volume available to each seedling is not predictive of the growth within the range considered, which could be due to the significantly shorter cycle observed than considered by Romano et al. [6]. Plus, in a DWC system the large majority of the root mass is dangling in the nutrient solution below the flat, so the cell volume available does not necessarily restrict root growth or nutrient uptake.

A custom-designed flat with half as many cells as the Speedling 338, but spaced twice as far apart would cut medium costs roughly in half. The relative costs of medium and seeds are such that compensating for the reduced germination associated with sowing more seeds in each cell (as was observed in Experiment 1) with 10% more seeds while using half the medium would result in a large proportional decrease in material input costs per flat. These preliminary tests corroborate similar results observed with the spinach cultivar Alrite and highlight the potential for a flat custom designed for baby spinach production [9,16,17].

Another promising observation from Experiment 1 was that the canopies of blocks in both treatments were full and relatively uniform. This validates the experimental design assumption that the canopy density effects between the treatments would be the same. It also suggests that seedlings will not be shaded out by neighbor plants when more seeds are planted in each cell if the overall canopy density is not too high and there is adequate room adjacent to the cell for seedlings to expand into during growth.

**Author Contributions:** D.B.J. and M.B.T. conceived and designed the experiment; D.B.J. performed the experiment; D.B.J. and M.B.T. analyzed the data; D.B.J. wrote the paper with significant contributions from M.B.T.

**Acknowledgments:** This research was supported by the Cornell University Department of Biological and Environmental Engineering as well as Element Farms, Inc. who supplied the seeds. We would like to thank Francoise Vermeylen from the Cornell Statistical Consulting Unit for her guidance on experimental design and statistical analysis. We express our gratitude to Olav Imsdahl (Cornell M.Eng 2017) for his work on 3D printed dibbling tools. Additional thanks to Alan Taylor and David deVilliers for advice and guidance and Neil Mattson for equipment and greenhouse space.

**Conflicts of Interest:** The authors declare no conflict of interest.

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
