# Peer review of "Effects of Seeding Pattern and Cultivar on Productivity of Baby Spinach (Spinacia oleracea) Grown Hydroponically in Deep-Water Culture"

_horticulturae, doi:10.3390/horticulturae5010020_

Round 1

Reviewer 1 Report

The manuscript submitted to horticulturae fit with the general scope of the Journal The quality is good but can be improved. However, I felt that the manuscript is a little weak from a scientific standpoint. First, if the method (seeding more seed per cell) the authors are trying to evaluate would lead to a negative outcome as they hypothesized (negative effect on germination and sprouting), one should wonder the practical importance of evaluating it. Also, for a manuscript which basically only evaluated two variables (sprout count and fresh weight), it is unnecessary too long due to extraneous details (lines 38-39, figure 1, lines 65-69, just to point out a few; the material and methods too long). Why values in each block were presented in table 3 and 4 which should just show the average and the P values. How is Figure 9 related to the objectives of this study?

The manuscript cannot be accepted for publication in its current form

Author Response

Reply to REviewer #1.

Excellent suggestions, all of which were incorporated. M&M was reduced. One table was eliminated. Clarifications made where suggested. FIgure 9 pertinence was addressed by text notation (important for reader to see an actual flat).

Tables were redone. Statistical methods were clarified.

Note that I started with making corretions from Reviewer #2 (had a PDF to work from).

Reviewer 2 Report

Please see attached document for comments and suggestions. Overall, I believe there is a foundation for a good manuscript. However, the authors need to edit, fix grammar and sentence structure, and wording. In addition, the statistics of the paper need to be improved. There was no indication of statistical design, replications, and how the data was analyzed. The results need to be reworked according to improvement in statistical analysis. The tables in the results are difficult to understand but can be improved with a new statistical approach. The presentation of the manuscript needs to be improved in order to be published. I would suggest for the authors to rewrite many of the sections of the introduction, results, and discussion and cut down on the materials and methods section. 

Author Response

REviewer 2's edits and suggestions were very helpful. All were incorporated. Once minor point is how units are expressed, e.g., 2 cm vs. 2cm. I believe currently we write units compressed, e.g., 2cm.

There was significant over-all rewrite to make the paper more understandable. For example, we changed use of Trial to Experiment, as Trial suggests a subsequent duplicate experiment.  There were two experiments, each addressing a different objective.  Within an experiment, multiple replicates were used (we called them blocks, but they are replications).

The stats were overseen by the head of the COrnell University statistical services department, so we're confident of the procedures used. Jump Pro was included to reference what stat package was used.

Thanks to this reviewer as he/she clearly read the paper closely.

Reviewer 3 Report

very well written and of great practical use by producers.

the title of the tables need to be rewritten to better describe the content of the table. the title should clearly and completely describe the table.

i didn't see an extensive discussion and comparison to other work done in this field.

Author Response

Reviewer #3.

Thank you for your excellent review and good suggestions. All have been addressed and incorporated. Thank you.

Reviewer 4 Report

The current study was conducted and the results interpreted properly. The authors have addressed limitations of the study and the results support their useful conclusions.

Author Response

Reviewer 4.

Thank you for reviewing. The other reviewers made many suggestions which I addressed and incorporated, e.g., retype tables, rewrite table titles, shorten M&M; better describe statistical methods.

MBT

Round 2

Reviewer 2 Report

Good job on the review. The manuscript reads much better. I do not have any further corrections. However, I do feel that the manuscript is lacking data tables/figures to display the data. I only came across one data table for the whole paper. Do the authors have any more data tables/figures they can add to the manuscript?

Author Response

Reviewer makes a good suggestion to add back more data into the manuscript to help explain the data results in the text. so, I added Table 3 which shows the Experiment 1 results.  Much clearer now to a reader. Particularly because the yield (fresh weight g) was NOT different between treatments, but was CLOSE. So reader can see this.